# Modulating Morphological and Redox/Glycative Alterations in the PCOS Uterus: Effects of Carnitines in PCOS Mice

**DOI:** 10.3390/biomedicines11020374

**Published:** 2023-01-27

**Authors:** Maria Grazia Palmerini, Guido Macchiarelli, Domenica Cocciolone, Ilaria Antenisca Mascitti, Martina Placidi, Teresa Vergara, Giovanna Di Emidio, Carla Tatone

**Affiliations:** Department of Life, Health and Experimental Sciences, University of L’Aquila, 67100 L’Aquila, Italy

**Keywords:** PCOS, DHEA, methylglyoxal, carnitines, oxidative stress, glycative stress, SIRT1, mitochondria, uterus, mouse

## Abstract

(1) Background: Polycystic ovarian syndrome (PCOS) is a common and multifactorial disease affecting reproductive-age women. Although PCOS ovarian and metabolic features have received extensive research, uterine dysfunction has been poorly investigated. This research aims to investigate morphological and molecular alterations in the PCOS uterus and search for modulating effects of different carnitine formulations. (2) Methods: CD1 mice were administered or not with dehydroepiandrosterone (DHEA, 6 mg/100 g body weight) for 20 days, alone or with 0.40 mg L-carnitine (LC) and 0.20 mg acetyl-L-carnitine (ALC) in the presence or absence of 0.08 mg propionyl-L-carnitine (PLC). Uterine horns from the four groups were subjected to histology, immunohistochemistry and immunoblotting analyses to evaluate their morphology, collagen deposition, autophagy and steroidogenesis. Oxidative-/methylglyoxal (MG)-dependent damage was investigated along with the effects on the mitochondria, SIRT1, SOD2, RAGE and GLO1 proteins. (3) Results: The PCOS uterus suffers from tissue and oxidative alterations associated with MG-AGE accumulation. LC-ALC administration alleviated PCOS uterine tissue alterations and molecular damage. The presence of PLC prevented fibrosis and maintained mitochondria content. (4) Conclusions: The present results provide evidence for oxidative and glycative damage as the main factors contributing to PCOS uterine alterations and include the uterus in the spectrum of action of carnitines on the PCOS phenotype.

## 1. Introduction

Polycystic ovarian syndrome (PCOS) is a common and multifactorial disease affecting 8–13% of reproductive-age women [1]. Clinical features for PCOS diagnosis include oligo-ovulation or anovulation, hyperandrogenism and polycystic ovarian morphology (PCOM) on ultrasound. According to the Rotterdam criteria, two out of the three features of PCOS are required for a diagnosis [2]. In 2009, the AE-PCOS Society established that for a diagnosis of PCOS, clinical androgenization or biochemical hyperandrogenism, or both, should be present [3]. Additionally, this syndrome is frequently linked to metabolic disorders, infertility, insulin resistance, and, in the long term, diabetes and cardiovascular disease [4].

Although it is the leading cause of anovulatory infertility, the pathogenesis of PCOS is still elusive. A crucial role is played by oxidative and glycative stresses. Increased levels of reactive oxygen species (ROS) promote the formation of AGEs (advanced glycation end-products) from glycative stress [5,6]. Numerous studies have hypothesized that AGEs can alter steroid hormone biosynthesis in the PCOS ovary, attacking enzymatic function and inducing changes in granulosa and theca cell functions, which are responsible for an inflammatory condition resulting in insulin resistance [7]. Many factors are involved in the development of oxidative stress, including increased adipose tissue, defects in mitochondrial metabolism, fatty acid oxidation and hyperglycemia [8]. 

The maintenance of the oxidative status and inflammatory state in PCOS is associated with a decrease in antioxidants. Several clinical studies reported that non-pregnant PCOS patients often exhibit decreased antioxidant concentrations in the ovarian granulosa cells and leukocytes [9,10]. Consistently, natural antioxidant supplementation leads to a significant improvement in menstrual cyclicity, acne and hirsutism in PCOS patients, with significant body weight and BMI reduction [11]; moreover, antioxidant therapy ameliorates ovarian follicle architecture and related hormone profile levels in PCOS models [12]. Then, natural antioxidants promote insulin sensibility and modulate the inflammatory response, helping to restore a functional balance in PCOS patients and animal models [12,13,14].

PCOS causes impairments to the endometrium’s phenotype and function associated with pregnancy loss, premature delivery, endometrial hyperplasia, and uterine carcinoma [15,16]. Endometrial dysfunctions are due to the concomitance of several complications, including hormonal and immune problems. In the PCOS endometrium, the number of natural killer cells, which are involved in the secretion of pre- and post-implantation angiogenic factors, is considerably reduced, leading to altered angiogenesis [17]. Consequently, the above-described alterations may hamper uterine receptivity in the “implantation window” [18]. The available literature reports that the human uterine peristalsis in a non-pregnant uterus is reduced in PCOS patients, with potential effects on sperm motility [19,20]. 

PCOS animal models have been essential for the investigation of PCOS uterine features, although data are still limited. The overall uterine thickness increased in a PCOS model induced by DHEA (dehydroepiandrosterone) in rats due to altered cell organization, collagen deposition and water absorption [21]. Endometrial hyperplasia, with enlarged endometrial glands with cystic dilatations, was shown in a PCOS model induced by letrozole in hamsters, with increased antioxidant enzymes, such as superoxide dismutase (SOD) and catalase (CAT), and altered metabolic parameters, such as increased leptin and insulin and decreased insulin receptor and glucose transporter 4 [22]. In DHEA mice, uterine receptivity and decidualization are impaired, with altered expression of implantation-related genes [23].

Potential treatments known to positively affect PCOS uterine dysfunction have been poorly investigated. Carnitine supplementation has been successfully employed in PCOS patients, with improvements in hormonal and metabolic parameters and ovarian function [24]. It is known that the serum concentration of total L-carnitine (LC) is reduced in PCOS patients, with a more pronounced effect in obese PCOS women [25,26]. Carnitine plays a fundamental role in fatty acid metabolism and is known to exert antioxidant effects through direct and indirect actions [27]. In a recent work from our research group, carnitine formulations containing LC and acetyl-L-carnitine (ALC) orally administered to DHEA (dehydroepiandrosterone)-induced PCOS mouse model were effective in ameliorating ovarian function. The addition of propyonyl-L-carnitine (PLC) also induced a decrease in serum testosterone [28]. 

Due to the limited available data on uterine alterations and underlying mechanisms in the PCOS phenotype, the aim of this research is to investigate morphological uterine PCOS features and the possible involvement of redox and glycative processes and to establish whether the administration of different carnitine formulations can alleviate these alterations. To this end, we employed a validated DHEA-induced mouse PCOS model showing a hormonal and ovarian PCOS phenotype [28,29] as well as defective uterus functionality [23].

## 2. Materials and Methods

### 2.1. Animals 

Outbred CD-1 mice (Charles River Italia s.r.l., Calco, Italy) were maintained in a temperature-controlled environment under a 12 h light/dark cycle (7.00–19.00) and given free access to feed and water ad libitum. All the experiments were carried out in conformity with national and international laws and policies (European Economic Community Council Directive 86/609, OJ 358, 1 12 December 1987; Italian Legislative Decree 116/92, Gazzetta Ufficiale della Repubblica Italiana n. 40, 18 February 1992; National Institutes of Health Guide for the Care and Use of Laboratory Animals, NIH publication no. 85-23, 1985). The project was approved by the Italian Ministry of Health and the internal Committee of the University of L’Aquila (Authorization n.269/2018-PR).

Four-week-old CD-1 female mice with a body weight of 20–21 g was randomly assigned to four groups (10 in each). To establish the PCOS model, mice were daily subcutaneously injected with DHEA (6 mg/100 g body weight, 100 μL/mouse in sesame oil with 10% of 95% ethanol, Sigma-Aldrich, St. Louis, CO, USA) for 20 consecutive days (DHEA group). The vehicle control group was injected with 0.09 mL sesame oil and 0.01 mL 95% ethanol daily for 20 consecutive days (control group). At the same time, mice received, by oral gavage, carnitine formulation 1 (0.40 mg LC, 0.20 mg ALC, DHEA/LC-ALC group) and carnitine formulation 2 (0.40 mg LC, 0.20 mg ALC, 0.08 mg PLC, DHEA/LC-ALC-PLC group) dissolved in water, 100 µL/mouse daily for 20 consecutive days. The control and DHEA groups received daily oral administration of water for 20 consecutive days. Mice were sacrificed by an inhalant overdose of carbon dioxide (CO_2_, 10–30%), followed by cervical dislocation. All efforts were made to minimize suffering. After opening the peritoneal cavity, ovaries, oviducts and uterine horns were removed from each side and transferred in PBS before further processing.

### 2.2. Estrous Cycle Determination

Daily vaginal smear analyses began the seventh day after the initial DHEA or vehicle injection. Estrous cycle stages were assessed using a wet smear method [30]. Saline lavage was used to obtain vaginal cells, which were subsequently examined under a light microscope with a 10× objective. The proestrus stage was marked by the predominance of nucleated epithelial cells and some cornified epithelial cells, the estrus stage by the predominance of cornified squamous epithelial cells, the metestrus stage by the predominance of leukocytes, and the diestrus stage by the predominance of leukocytes.

### 2.3. H & E Staining and Morphometric Analysis

After washing in PBS, uterine horns were fixed in 3.7% paraformaldehyde (PFA) in PBS (Bio-Optica, Milan, Italy) for 12–16 h, carefully washed in PBS, dehydrated in an ascending series of alcohol, clarified in xylene and embedded in paraffin blocks. Sections were cut with a microtome (Leica SMR2000, Wetzlar, Germany) and sliced into 6 µm serial sections. Sections were then deparaffined and hydrated through xylenes and descending series of alcohol, stained with H&E according to the manufacturer’s instruction (Bio Optica, Milan, Italy) and observed by light microscopy (Zeiss Axiostar Plus, Oberkochen, Germany). The luminal epithelial cell height was measured from the apical surface to the basement membrane, and endometrial thickness (height) measurements were investigated according to [31] using ImageJ software (http://rsbweb.nih.gov/ij/, accessed on 23 November 2022). Statistical analyses were performed. All experiments were repeated at least three times, and data were expressed as mean ± SD. Statistical comparisons were performed using one-way ANOVA with Tukey’s HSD tests for post hoc analysis (ezAnova).

### 2.4. Mallory Trichrome Staining

Paraffin-embedded sections of formalin-fixed uterus tissue were deparaffinized and hydrated through xylenes and graduated alcohol series and processed for Mallory Trichome Staining (Bio Optica, Milan, Italy) according to the manufacturer’s instructions.

### 2.5. Immunohistochemical Analysis: MG-AGE and 4-HNE

Paraffin-embedded sections of formalin-fixed uterine horns were deparaffinized and hydrated through xylenes and graded alcohol series. To increase the immunoreactivity, sections were boiled in 10 mM citrate buffer (pH: 6.1. Bio-Optica, Milan, Italy) in a microwave at 720 W (3 cycles/3 min each). Then, sections were subjected to treatment for blocking endogenous peroxidase activity (Dako). After thorough washing, sections were incubated with M.O.M. mouse IgG blocking reagent overnight at 4 °C (Vector Laboratories Burlingame, CA, USA,) according to the manufacturer’s protocol. Sections were then incubated with mouse monoclonal to methylglyoxal (MG)-AGE (Arg-Pyridine, AGE06B, BioLogo, Kronshagen, Germany, 1:100) antibody or rabbit polyclonal to 4-HNE (4 Hydroxynonenal, ab46545, Abcam, Cambridge, UK, 1:100) diluted in M.O.M. diluent for 40 min, according to the Vector Laboratories instructions. MG-AGE and 4-HNE were revealed by Labelled Polymer-HRP, 3,3-diaminobenzidine (DAB) substrate buffer and DAB (Dako, Glostrup, Denmark), according to the manufacturer’s instructions. Counterstaining was performed with hematoxylin (Bio-Optica, Milan, Italy). Negative control was performed by omitting the primary antibody and substituting it with M.O.M. diluent alone. Sections were dehydrated, mounted with Neomount (Merck, Darmstadt, Germany), observed and photographed under a Leiz Laborlux S microscope (Oberkochen, Germany) equipped with an Olympus digital compact camera. Evaluation and automated scoring of immunohistochemistry (IHC) signals were performed using Image J 1.44p software (IHC profiler plugin), according to [32].

### 2.6. Immunohistochemical Analysis

Paraffin-embedded sections of formalin-fixed uterine horns were deparaffinized and hydrated through xylenes and graded alcohol series. To increase the immunoreactivity, the sections were boiled in 10 mM citrate buffer (pH 6.1, Bio-Optica, Milan, Italy) in a microwave at 720 W (3 cycles/3 min each). Then sections were subjected to treatment for blocking endogenous peroxidase activity (Dako, Glostrup, Denmark). After thorough washing, sections were incubated with 5% BSA/PBS (Sigma-Aldrich, St. Louis, CO, USA) for 1 h at RT. Then, uterus sections were incubated with the following primary antibodies: rabbit polyclonal to 17beta-hydroxysteroid dehydrogenase type 4 (17 β-HSD4, 1:100), translocase of outer mitochondrial membrane 20 (TOMM20, 1:400) (Thermo Fisher Scientific, Rockford, IL, USA) and rabbit polyclonal to COL1 (1:500) (Ab-82138, Immunological Sciences, Rome, Italy), all diluted in 1% BSA/PBS overnight at 4 °C. After washing with PBS, rabbit antibodies were revealed by labelled polymer-HRP, 3,3-diaminobenzidine (DAB) substrate buffer and DAB (DAKO, Glostrup, Denmark), according to the manufacturer’s instructions. Counterstaining was performed with hematoxylin (Bio-Optica). A negative control was performed by omitting the primary antibody and substituting it with 1% BSA/PBS diluent alone. Sections were dehydrated and mounted with Neomount (Merck, Darmstadt, Germany). They were observed and photographed under a Leiz Laborlux S microscope (Oberkochen, Germany) equipped with an Olympus digital compact camera (Hamburg, Germany). The evaluation and automated scoring of immunohistochemistry (IHC) signals was performed as described above.

### 2.7. Western Blot Analysis

Part of the uterus stored at −80 °C was processed for protein extraction. Uterine tissues were homogenized in RIPA buffer by repeated freeze/thaw cycles in liquid nitrogen. After centrifugation (14,000 rpm for 90 min at 4 °C), the supernatants were collected for protein analysis. Protein concentration was determined by a BCA protein assay kit (Pierce, Rockford, IL, USA). Protein samples were separated by SDS-PAGE and transferred to a polyvinylidene difluoride membrane (Sigma-Aldrich, St. Louis, MO, USA). Non-specific binding sites were blocked for 1 h at room temperature with 5% non-fat dry milk (Bio-Rad Laboratories, Segrate, Italy) in Tris-buffered saline containing 0.05% Tween 20 (TBS-T). Membranes were incubated with polyclonal rabbit anti-SIRT1 antibody (Ab12193, Abcam, Cambridge, UK; 1:700), anti-SOD2 antibody (Ab86087, Abcam, Cambridge, UK; 1:1000), anti-GLO1 antibody (MA1-13029, Thermo Fisher Scientific, Waltham, MA, USA; 1:400), anti-PGC1α antibody (SC-13067, Santa Cruz Biotechnology Inc., Dallas, TX, USA, 1:500), anti-RAGE (PA1-075, ThermoFisher Scientific, Waltham, MA, USA, 1:750), anti-LC3A/B antibody (AB-83557, Immunological Sciences, Rome, Italy, 1:500), anti-P62 antibody (AB-83779, Immunological Sciences, Rome, Italy, 1:500) or anti-GAPDH (TA802519, OriGene Technologies Inc., Rockville, MD, USA, 1:750) overnight at 4°C, followed by incubation with horseradish peroxidase (HRP) conjugated anti-rabbit (BA1054, Boster Biological Technology Co., Ltd., Pleasanton, CA, USA, 1:3000) or anti-mouse secondary antibody (Ab6728, Abcam, Cambridge, UK, 1:2000) for 1 h at room temperature. After washing, specific immunoreactive complexes were detected by an ECL kit (Thermo Fisher Scientific, Waltham, MA, USA) and a Uvitec Cambridge system (Alliance series, Cambridge, UK). The bands were normalized for GAPDH using ImageJ 1.44p software, and values were given as relative units (RU). The experiment was performed in triplicate.

### 2.8. Statistical Analysis 

Data processing and statistical analyses were performed using Excel (2022, Microsoft) and GraphPad Prism 8 (GraphPad software, Inc., San Diego, CA, USA) by unpaired *t*-test after checking for a normal distribution. All experiments were performed with a minimum of three biological replicates from each experimental group, as reported in figure legends. All data are presented as mean ± SEM). *p*-values < 0.05 were considered statistically significant. 

## 3. Results

This study relied on a PCOS mouse model generated by DHEA administration in the presence or in the absence of two different carnitine formulations, according to previous research [28,29]. Following the determination of the PCOS phenotype in terms of estrous cycle alteration, experiments were carried out to characterize morpho-functional and molecular alterations in the PCOS uterus and the possible effects of carnitine treatments. To this end, we searched for DHEA-related alterations and carnitine-associated effects on uterus morphology, collagen deposition, autophagy and steroidogenesis. The redox-related profile and the oxidative-/MG-dependent molecular damage and underlying signaling were investigated.

### 3.1. PCOS Phenotype

An analysis of estrous cyclicity was conducted to evaluate the efficacy of DHEA administration. In contrast to the control group, DHEA mice showed abnormal estrous cyclicity. The representative cyclicity of the mice of the two groups is shown in Figure 1.

### 3.2. Uterine Hyperplasia and Fibrosis 

Histological analysis showed in all groups a well-preserved microscopic organization of the murine uterine horns that, in cross-sections and from the luminal side, consisted of a tunica mucosa, muscularis and serosa. In the control group, the HE staining showed an endometrium with a simple columnar epithelium, made by non-ciliated secretory cells and less-represented ciliated cells, and an underlying stroma with a highly cellular lamina propria hosting single tubular uterine glands. The mean width of the epithelium and endometrium was 12.464 ± 1.685 and 141.615 ± 13.266 µm (Table 1). The myometrium was constituted by three muscular layers, with differently orientated muscular fibers and the perimetrium by a thin layer of mesothelium and loose connective tissue (Figure 2A–D). In the DHEA group, the endometrium appeared hyperplastic and thicker than in the controls; the luminal epithelium thickness increased and was invaginated in the underlying stroma, which showed lower cellularity compared to the control group (Figure 2E–H). The mean width of the epithelium and endometrium in the DHEA group was, respectively, 25.119 ± 2.370 and 345.584 ± 17.199 µm (Table 1).

In the DHEA/LC-ALC group, the general histological architecture was more similar to controls (Figure 2I–L). The morphometric analysis of the epithelium did not evidence significative differences with respect to the control and DHEA groups (epithelium: 19.435 ± 2.582; *p* > 0.05); differently, the endometrium significantly decreased with respect to the DHEA group (237.225 ± 40.884 µm vs. 345.584 ± 17.199 µm, DHEA/LC-ALC vs. DHEA; *p* < 0.01) but not with respect to the control (237.225 ± 40.884 µm vs. 141.615 ± 13.266 µm; *p* < 0.01) (Table 1). The DHEA/LC-ALC-PLC group was characterized by a thinner endometrial epithelium, although this was not homogeneous, and a reduced thickness of the endometrial wall (respectively, 13.747 ± 1.494 µm and 171.683 ± 8.181 µm), which was significantly different with respect to DHEA (*p* < 0.01) and with a situation more similar to control group (Figure 2M–P, Table 1). 

In the DHEA group, hyperplasia was found both in the glandular (Figure 2F) and luminal epithelium (Figure 2G), with the formation of multiple cell layers, higher vascularity in the endometrial stroma (Figure 3A), and an increased presence of inflammatory cells (eosinophils, lymphocytes, and macrophages) (Figure 3B). Moreover, the number of uterine glands decreased, as seen in cross-sections, but they showed multiple glandular dilatations with epithelial hyperplasia and a cellular disposition less regular than in the controls (Figure 2E,F). In both groups treated with carnitines, uterine glands decreased in number and volume if compared to the DHEA and control groups, in association with a reduced luminal and glandular epithelium hyperplasia and a diminished presence of inflammatory stromal cells (Figure 2I,J,M,N). The DHEA group also was characterized by uterine hyperfibrosis compared to the control group, as demonstrated by an increased collagen deposition in the lamina propria (Figure 4E) and muscle fibrosis (Figure 4G and inset) after Azan Mallory and type 1 collagen (Col1) staining. A reduction of the fibrotic tissue was seen in DHEA/LC-ALC and DHEA/LC-ALC-PLC groups, even if more evident in the former (Figure 4I–O).

### 3.3. Autophagic Flux

Autophagy is known to drive essential homeostatic mechanisms in uterus physiology and has been involved in uterus hyperplasia [33]. To quantify the level of autophagy, Western blotting was used to detect the expression levels of LC3II/LC3I and p62, the two of which can be used to evaluate the status of autophagy flux [34]. The results showed that the LC3II/LC3I ratio in the DHEA group decreased significantly, while p62 increased (Figure 5), indicating the decreased level of autophagy in the DHEA uterus. The presence of carnitine modulated the LC3II/LC3I ratio, which increased to levels higher than in the control. The p62 amount in the carnitine groups was maintained at a level not different from DHEA and increased.

### 3.4. Steroidogenesis

Based on previous data about DHEA-induced alterations in 17 β-HSD4 expression in the mouse ovary [28,29], in this study, we evaluated uterine expression and spatial distribution of this steroidogenic enzyme that inactivates estradiol by conversion to estrone in the porcine uterus and converts androstenedione to DHEA [35]. Immunohistochemical analysis of 17 β-HSD4, mainly expressed in the luminal and glandular epithelium and in the stroma of controls (Figure 6A), revealed an increased expression in all the endometrial compartments, but also in the myometrium of the DHEA group (Figure 6B). An expression pattern more comparable to controls was noticed in the DHEA/LC-ALC group, especially for the glandular epithelium (Figure 6C); in the DHEA/LC-ALC-PLC group, the stromal and glandular epithelium expression of 17 β-HSD4 appeared less intense (Figure 6D).

### 3.5. Oxidative Damage and Signaling 

Oxidative damage was investigated by examining the expression pattern of 4-hydroxynonenal (HNE) protein adducts, well-known markers of lipid peroxidation and oxidative stress damage [36]. The analysis has revealed high 4-HNE staining in the epithelial cells of the control endometrium (Figure 7A). The increased immunostaining for 4-HNE in the luminal and glandular endometrial epithelium, as well as in the myometrium found in the DHEA group, was reduced in the DHEA/LC-ALC group (Figure 7B,C). In the DHEA/LC-ALC-PLC group (Figure 7D), a decrease in the HNE staining was observed in the lamina propria and myometrium.

Considering the role of SIRT1 as a regulator of oxidative stress signalling [37], we investigated the level of expression of this protein to understand whether it is involved in the oxidative response in the DHEA uterus. Western blot analysis of SIRT1 revealed that its expression increased in the DHEA uterus when compared with the control group. In mice exposed to carnitine during DHEA treatment, SIRT1 levels were found to be lower with a more consistent effect in the formulation including PLC (Figure 8). Then, the level of the expression of SOD2, known to be activated downstream to SIRT1, was monitored in the experimental group. As shown in Figure 7, SOD2 protein was significantly upregulated in the DHEA uterus in comparison with untreated mice. Both carnitine formulations did affect this change. 

### 3.6. Mitochondrial Damage and Signaling

Effects on mitochondria were investigated by evaluating TOMM20 staining and protein expression of PGC1*α*, the main regulator of mitochondrial biogenesis and function [38,39]. Immunohistochemical analysis of the mitochondrial protein TOMM20 in controls showed intense staining in the apical domains of the luminal and glandular epithelium of the endometrium as well as in the myometrial muscle cells (Figure 9A). A reduced expression of the mitochondrial transporter was observed in the DHEA group with respect to controls (Figure 9B). An expression pattern similar to controls was found after carnitine administration, even if it was less intense in the DHEA/LC-ALC group (Figure 9C) and intensified in the DHEA/LC-ALC-PLC group (Figure 9D). Western blot analysis of PGC1*α* revealed a significant increase in DHEA irrespective of carnitine administration (Figure 10).

### 3.7. Glycative Damage and Signalling

Uterine horns from controls showed MG-AGE staining mainly located in the cytoplasm of the glandular and luminal epithelial cells, with a slight diffusion in the endometrial stroma (Figure 11A and inset). Differently, the DHEA group showed intense staining for MG-AGE levels, higher than in controls, also distributed in the tunica muscularis (Figure 11B). The two carnitine formulations induced different results, with a reduced expression of MG-AGEs in the DHEA/LC-ALC group (Figure 11C) and no evident effects with respect to DHEA in the DHEA/LC-ALC-PLC group (Figure 11D). Western blot analysis of RAGEs showed a significant decrease in DHEA mice and maintained the same level of expression in the two carnitine groups. The expression of GLO1 was found to be unchanged when all the groups were compared (Figure 12).

## 4. Discussion

This study described morphological and molecular changes in the uterine horns in a validated mouse model of PCOS induced by DHEA and the action of two different formulations of carnitines. The present results provide evidence that the PCOS uterus suffers from tissue and oxidative alterations associated with MG-AGE accumulation and that LC, ALC and PLC administration alleviates PCOS uterine tissue alterations and molecular damage with differential effects. 

In this study, we showed morphological evidence for increased proliferation in the uterine glandular epithelia, which may be responsible for the enlarged glands and thickened uterine walls [40,41]. Data also evidenced luminal and glandular epithelial hyperplasia, with an increased number of single tubular uterine glands and decreased uterine glands, which may represent critical factors in the reduced endometrial receptivity of PCOS patients [42,43]. A high level of estrogen is known to shorten the implantation window [44]. In DHEA mice, estrogen level undergoes a consistent increase, and in pregnant mice, the excess of androgen causes a decrease in serum progesterone level and an increase in serum estradiol level [23]. In the PCOS ovary, progesterone deficiency or progesterone resistance are responsible for endometrial hormonal imbalance, resulting in impaired endometrial function and increased risk of endometrial cancer [45]. Moreover, excessive androgen levels in PCOS increase the bioavailability of estrogens due to the peripheral conversion of androgen into estrogen, which in turn, upregulates the endometrial androgen receptor [46]. In this study, we do not present direct evidence for hormonal changes. Nevertheless, 17β-HSD4, the enzyme involved in the conversion of androstenediol into DHEA [35], was found to be upregulated as evidence of hormonal imbalance in the DHEA uterus. 

Increased collagen deposition represents a further phenotype of DHEA uterus, here revealed by Mallory trichome staining and COL1 expression. Elevated levels of type I collagen characterize fibrotic disorders in the reproductive tract [47]. In endometriosis, this condition has been related to the aberrant transcriptional activation of *Col1a1* and *Col1a2* genes, proving to be critical factors for endometrial receptivity [48]. 

Although the relationship between inflammation and PCOS is still under investigation [49], an increased number of eosinophils has been taken as evidence of inflammation in the DHEA group, probably as a consequence of hyperandrogenism. Inflammation may impair the physiological clearance of senescent decidual cells and turn into a chronic process resulting in defective functionality [50]. Inflammatory elements observed in the DHEA group may promote DHEA-related muscle fibrosis [40,51], a condition known to affect contractility and peristalsis by interfering with sperm and embryo transport and the maintenance of early pregnancy [52]. Overall, based on these observations, the PCOS uterus phenotype may resemble the aging uterine phenotype as described in preclinical models [50]. Muscle fibrosis disrupts uterine architecture in PCOS patients, predisposing them to uterine pathologies, such as endometrial cancer [21]. 

Carnitine supplementation has been successfully employed in PCOS patients with improvements in hormonal and metabolic parameters, increased energy consumption, and lipid and bodyweight reduction, but their mechanistic action is still under investigation. The endogenous carnitine pool is formed by the short-chain carnitine esters acetyl-L-carnitine (ALC) and propionyl-L-carnitine (PLC). When exogenously administered, ALC and PLC have a higher bioavailability in comparison to L-carnitine (LC) [24]. Our previous study in DHEA mice showed that the oral administration of LC and ALC alleviates ovarian dysfunction associated with PCOS and that the coadministration of PLC provides better activity [28]. Here, we found that treatment with both carnitine supplements reduced the abnormal epithelial hyperplasia found both in the luminal and glandular epithelium, hypertrophy, inflammation, fibrosis, and glandular alteration related to DHEA. Nevertheless, the absence of PLC in the carnitine formulation exerts a greater effect on uterine phenotype, except for endometrial thickness, which was counteracted by PLC more efficiently than LC-ALC. 

In consideration of its role in uterine dysfunctions [33], reduced autophagy has been proposed in this study as a factor contributing to DHEA-induced hyperplasia. However, based on the LC3II/LCI ratio and p62 level, beneficial carnitine actions are not ascribed to autophagy modulation.

An important finding of this study is the observation of oxidative molecular damage in the PCOS uterus, evidenced by an increased level of proteins with 4-HNE adducts. We observed that the administration of ALC-LC alleviated oxidative stress in all compartments, and the presence of PLC reduced beneficial effects in some DHEA uterine regions. In addition, oxidative damage in the DHEA uterus was associated with the activation of antioxidant signaling involving the upregulation of SIRT1 and SOD2. These two enzymes are part of the same network: the upregulation of SIRT1 orchestrates an adaptive response to mild oxidative stress and leads to the deacetylation of FOXO3A, known to activate the SOD2 gene [37]. Moreover, reduced mitochondrial content indicates that the DHEA-related redox alterations observed here may be related to mitochondria damage. Although no significant changes affected PGC1α, the main regulator of mitochondrial biogenesis and function [53], the role of the mitochondria is supported by the observation that mitochondrial content remained unchanged when the LC-ALC-PLC formulation was administered, along with SIRT1 levels, indicating a specific effect of PLC. SOD2 modulation appears to not be implicated in the beneficial action of PLC as evidence for multiple pathways underlying this regulation.

A relevant finding of this research is the demonstration that glycative stress is involved in uterine dysfunction associated with PCOS supporting the crucial role of AGE in this syndrome [6]. Recent reports state that AGEs are involved in excessive androgen production in PCOS because they modulate the activities of crucial steroidogenesis enzymes [54,55]. AGEs may interfere with intracellular insulin signaling and potentially affect ovarian function and follicular growth, contributing to anovulation [56]. Moreover, AGEs are actively involved in a positive feedback loop, leading to oxidative stress [57]. Oxidative stress promotes the last step of advanced glycation, hence, accelerating AGE accumulation. AGEs, in turn, activate pro-inflammatory pathways and reactive ROS generation [57]. One of the most powerful AGE precursors is MG, an α-oxo aldehyde derived from glucose autoxidation, lipid peroxidation, and the polyol pathway [58]. In addition to its activity as an AGE precursor, MG has deleterious effects on mitochondrial respiration, proliferation, survival, and redox balance [58]. The MG detoxification system includes two enzymes working in tandem: glyoxalase 1 (GLO1) and glyoxalase 2 (GLO2) [59]. Not widespread in the uterine stroma, MG-AGE deposition increased in DHEA mice in all uterine compartments, and their accumulation was reduced by the supplementation of LC-ALC, whereas the presence of PLC seemed to inhibit this effect. In contrast to the PCOS ovary and most tissues with high AGEs [28], in the PCOS uterus, we found a decreased expression of RAGEs, the membrane receptor for AGEs [60]. As previously reported [61], this condition may be ascribed to oxidative-dependent cleavage of RAGE to produce a soluble RAGE, cRAGE, a decoy receptor binding AGEs to avoid their interaction with RAGE triggering pro-oxidant and pro-inflammatory pathways. Carnitine administration did not change this molecular effect, indicating that other signaling is involved in the attenuation of AGE deposition observed in LC-ALC/DHEA mice. Likewise, no changes in the expression of GLO1 were observed, probably as evidence that MG-AGE accumulation in the DHEA uterus is more likely ascribed to increased oxidation rather than to increased MG levels.

## 5. Conclusions

Based on a thorough histological investigation and analysis of collagen deposition, autophagy and steroidogenesis, this study provides new information about PCOS morphological alterations at the uterine level. In addition, it reveals for the first time that, similarly to the ovary, the PCOS uterus suffers from oxidative damage associated with a glycative component in terms of AGE accumulation, mitochondrial damage and SIRT1-SOD2 axis upregulation. Importantly, the study provides strong evidence of differential effects of carnitines on uterine PCOS features: LC-ALC alleviates PCOS tissue alterations and molecular damage, while the addition of PLC protects against oxidative damage with tissue specificity but exhibits greater efficacy in preventing fibrosis, mitochondrial damage and maintaining SIRT1 homeostasis. These discrepancies may emerge from differences among LC, ALC and PLC in their pharmacokinetics, antioxidant and mitochondrial activity and lipid metabolism [62], an important aspect of uterine function [63]. Carnitine plays an essential role in mitochondrial fatty acid β-oxidation as a part of a cycle that transfers long-chain fatty acids to mitochondria and also to peroxisomes [64]. Although we have not specifically addressed this point, we reported here that the administration of a carnitine formulation containing PLC prevents 17 β-HSD IV upregulation induced by DHEA. Considering the additional role of this enzyme in peroxisomal fatty acid β-oxidation, the beneficial effects of carnitines on DHEA may be mediated by the modulation of this process. 

Overall, the present research supports the validity of DHEA mice for the study of PCOS uterine modifications and could open new frontiers in the search for prognostic markers and therapeutical strategies to counteract PCOS.

## Figures and Tables

**Figure 1 biomedicines-11-00374-f001:**
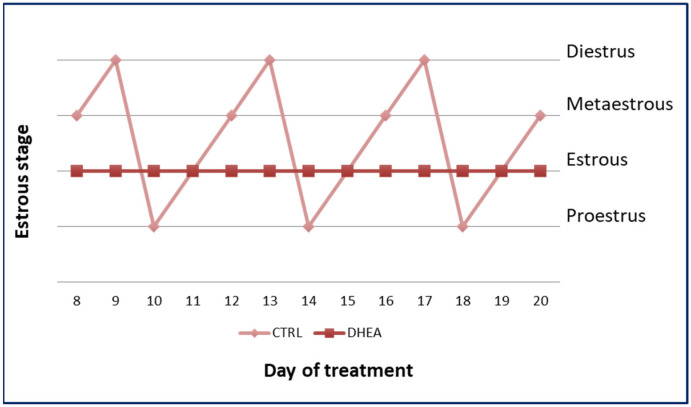
Estrous cycle in control and DHEA mice.

**Figure 2 biomedicines-11-00374-f002:**
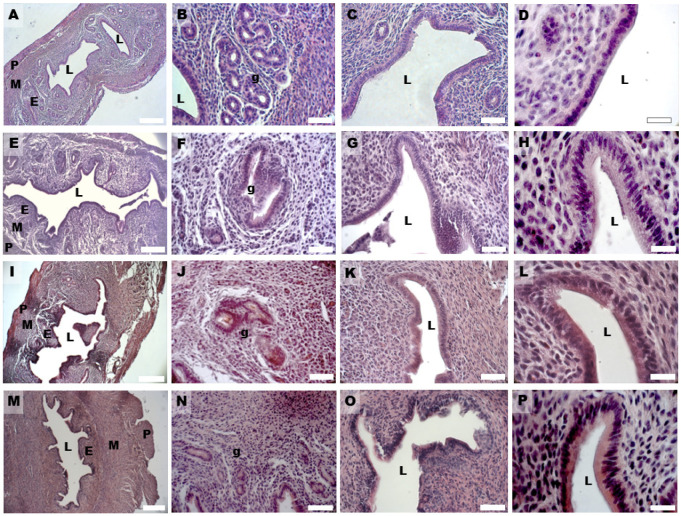
HE staining of mouse uterine horns of control (**A**–**D**), DHEA (**E**–**H**), DHEA/LC-ALC (**I**–**L**) and DHEA/LC-ALC-PLC (**M**–**P**) groups. (**A**,**E**,**I**,**M**) low-magnification LM pictures of the lumen (L), endometrium (E), myometrium (M) and perimetrium (P). (**B**,**F**,**J**,**N**) glandular epithelium (g). (**C**,**G**,**K**,**O**) luminal epithelium; (**D**,**H**,**L**,**P**) high magnification of the luminal epithelium made of columnar cells. LM, mag. 10× ((**A**,**E**,**I**,**M**) Bar: 200 µm), 20× ((**B**,**C**,**F**,**G**,**J**,**K**,**N**,**O**) Bar: 50 µm), 40× ((**D**,**H**,**L**,**P**) Bar: 20 µm).

**Figure 3 biomedicines-11-00374-f003:**
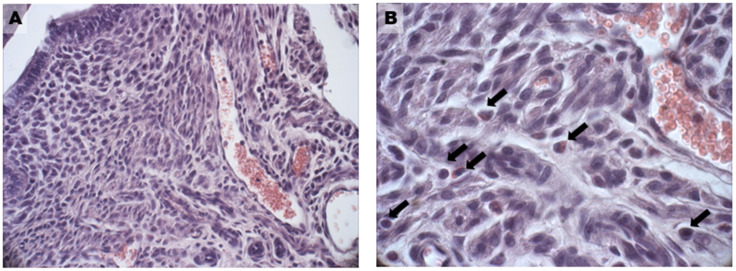
Abundant vascularization (**A**) and presence of inflammatory cells (**B**) in the endometrium of the DHEA group. Black arrows in (**B**) indicate single inflammatory cells as eosinophils, lymphocytes and macrophages. LM, 20× (**A**), 40× (**B**).

**Figure 4 biomedicines-11-00374-f004:**
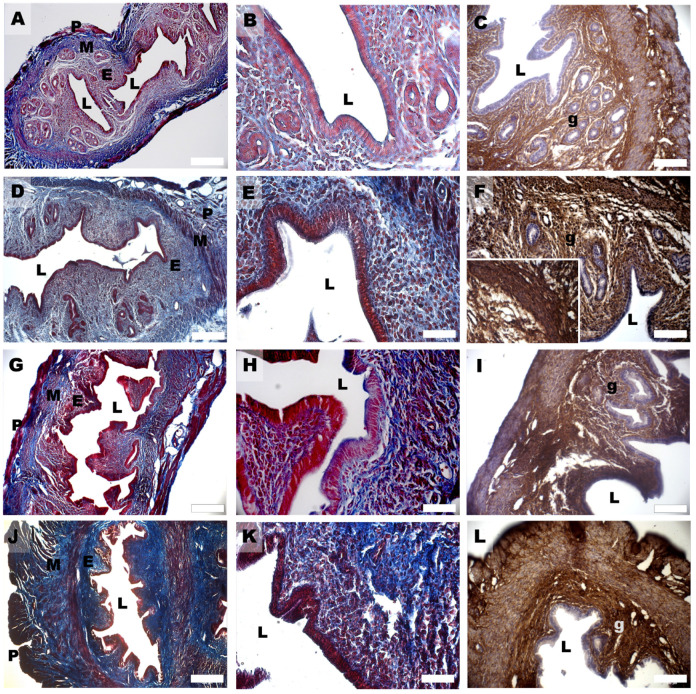
Mallory Trichrome (**A**,**B**,**D**,**E**,**G**,**H**,**J**,**K**) and Col1 (**C**,**F**,**I**,**L**) staining of mouse uterine horns of control (**A**–**C**), DHEA (**D**–**F**), DHEA/LC-ALC (**G**–**I**) and DHEA/LC-ALC-PLC (**J**–**L**) groups. (**A**,**D**,**G**,**J**) low-magnification LM pictures of the lumen (L), endometrium (E), myometrium (M) and perimetrium (P). (**B**,**E**,**H**,**K**) high magnification of the luminal epithelium. (**C**,**F**,**I**,**L**) glandular epithelium (g). Inset in F shows a detail of the tunica muscularis. LM, mag. 10× ((**A**,**D**,**G**,**J**) Bar: 200 µm), 20× ((**B**,**C**,**E**,**F**,**H**,**I**,**K**,**L**) Bar: 100 µm).

**Figure 5 biomedicines-11-00374-f005:**
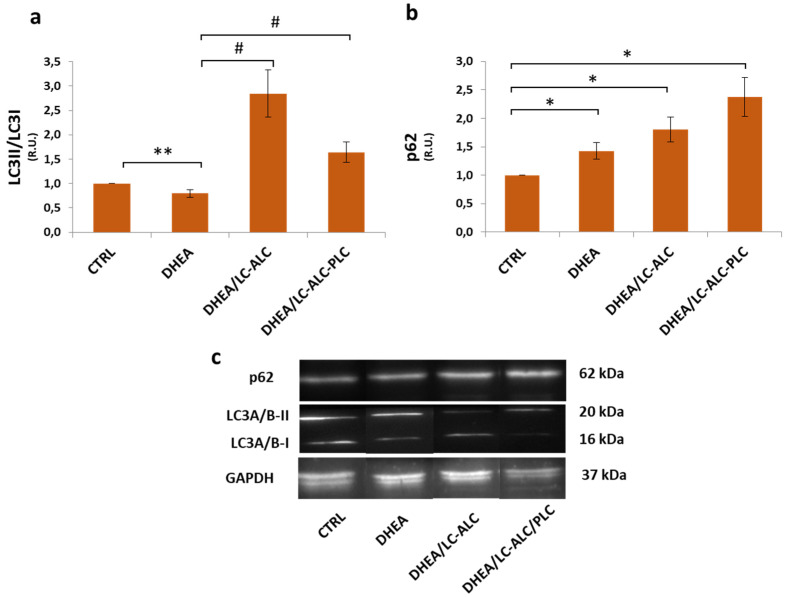
Effects of carnitine administration on DHEA-induced autophagy in mouse uterine horns. LC3II/LC3I ratio (**a**) and p62 (**b**) Western blotting analysis and representative images (**c**). Three mice per experimental group were employed. Experiments were conducted in triplicate. Data obtained from n = 9 observations are presented as means ± SEM of densitometric analysis of immunoreactive bands normalized to internal reference protein (glyceraldehyde-3-phosphate dehydrogenase, GAPDH). * *p* < 0.01 vs. CTRL, ** *p* < 0.05, # *p* < 0.01, *t*-test.

**Figure 6 biomedicines-11-00374-f006:**
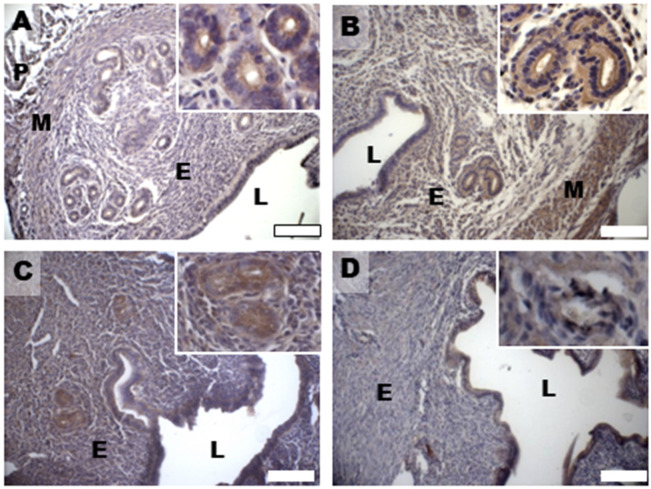
Immunostaining of mouse uterine horns for 17 β-HSD4 (**A**–**D**) of control (**A**), DHEA (**B**), DHEA/LC-ALC (**C**) and DHEA/LC-ALC-PLC (**D**) groups. Insets show details of the glandular epithelium. L: lumen; E: endometrium; M: myometrium; P: perimetrium. LM, mag. 10×. Insets: 20×.

**Figure 7 biomedicines-11-00374-f007:**
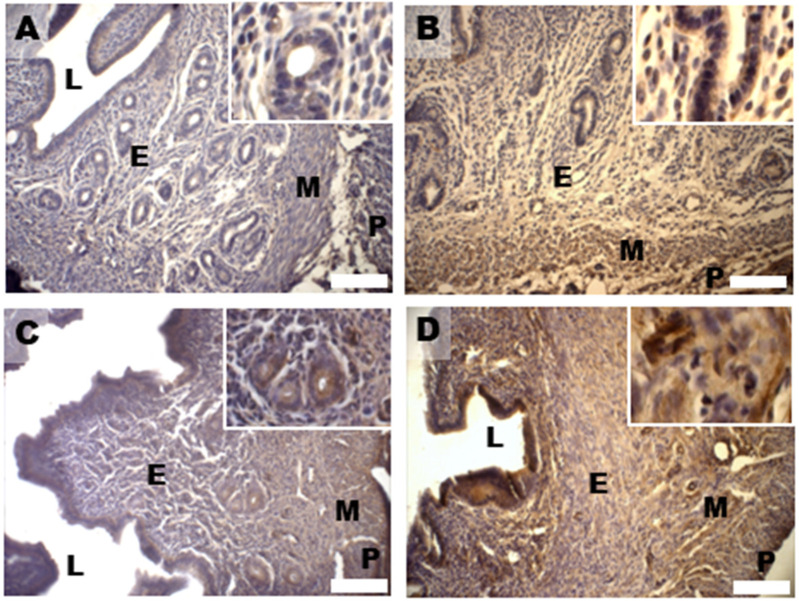
Immunostaining of mouse uterine horns for 4-HNE (**A**–**D**) of control (**A**), DHEA (**B**), DHEA/LC-ALC (**C**) and DHEA/LC-ALC-PLC (**D**) groups. Insets show details of the glandular epithelium. L: lumen; E: endometrium; M: myometrium; P: perimetrium. LM, mag. 10×. Insets: 20×.

**Figure 8 biomedicines-11-00374-f008:**
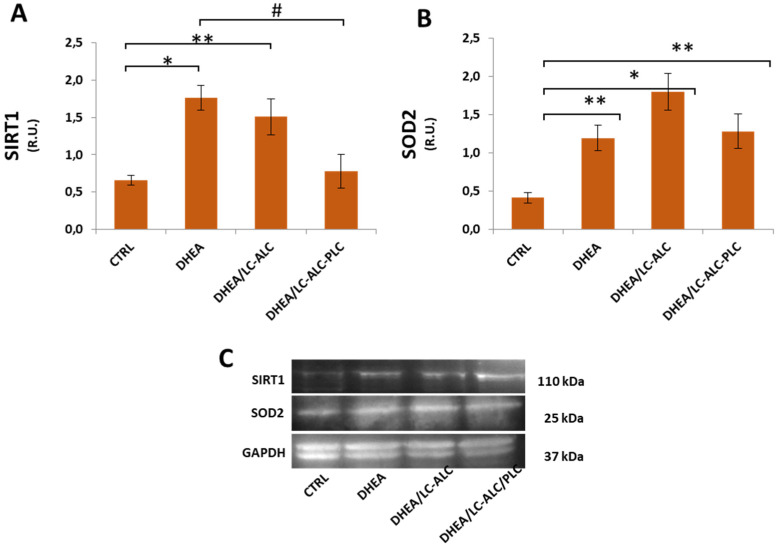
Western blot analysis of SIRT1 (**A**) and SOD2 (**B**) and representative images (**C**). Three mice per experimental group were employed. Experiments were conducted in triplicate. Data obtained from n = 9 observations are presented as means ± SEM of densitometric analysis of immunoreactive bands normalized to internal reference protein (glyceraldehyde-3-phosphate dehydrogenase, GAPDH). * *p* < 0.01 vs. CTRL, ** *p* < 0.05, # *p* < 0.01, *t*-test.

**Figure 9 biomedicines-11-00374-f009:**
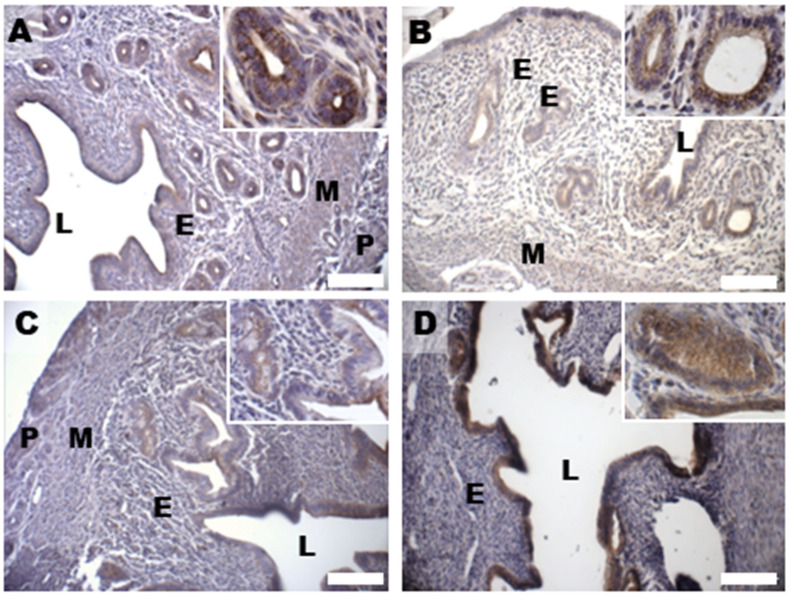
Immunostaining of mouse uterine horns for TOMM20 (**A**–**D**) of control (**A**), DHEA (**B**), DHEA/LC-ALC (**C**) and DHEA/LC-ALC-PLC (**D**) groups. Insets show details of the glandular epithelium. L: lumen; E: endometrium; M: myometrium; P: perimetrium. LM, mag. 10×. Insets: 20×.

**Figure 10 biomedicines-11-00374-f010:**
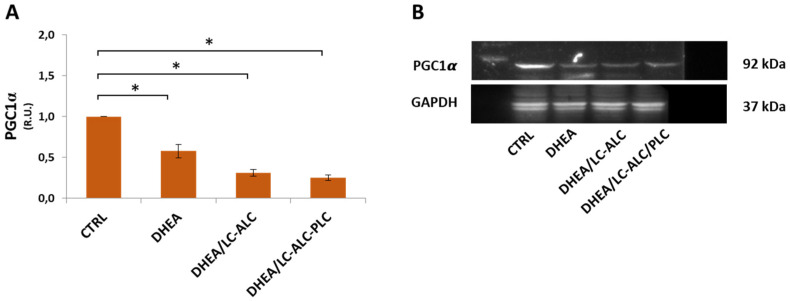
Western blot analysis of PGC1*α* (**A**) and representative images (**B**). Three mice per experimental group were employed. Experiments were conducted in triplicate. Data obtained from n = 9 observations are presented as means ± SEM of densitometric analysis of immunoreactive bands normalized to internal reference protein (glyceraldehyde-3-phosphate dehydrogenase, GAPDH). * *p* < 0.01 vs. CTRL, *t*-test.

**Figure 11 biomedicines-11-00374-f011:**
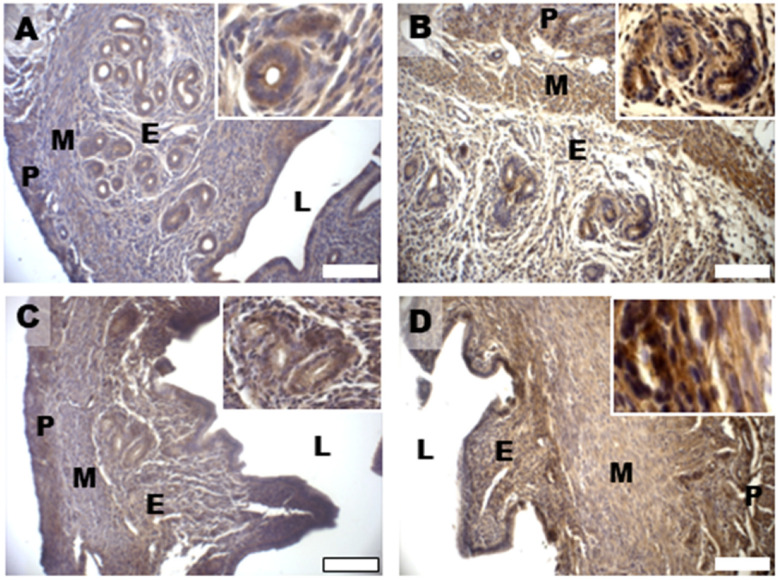
Immunostaining of mouse uterine horns for MG-AGE (**A**–**D**) of control (**A**), DHEA (**B**), DHEA/LC-ALC (**C**) and DHEA/LC-ALC-PLC (**D**) groups. Insets show details of the glandular epithelium. L: lumen; E: endometrium; M: myometrium; P: perimetrium. LM, mag. 10×. Insets: 20×.

**Figure 12 biomedicines-11-00374-f012:**
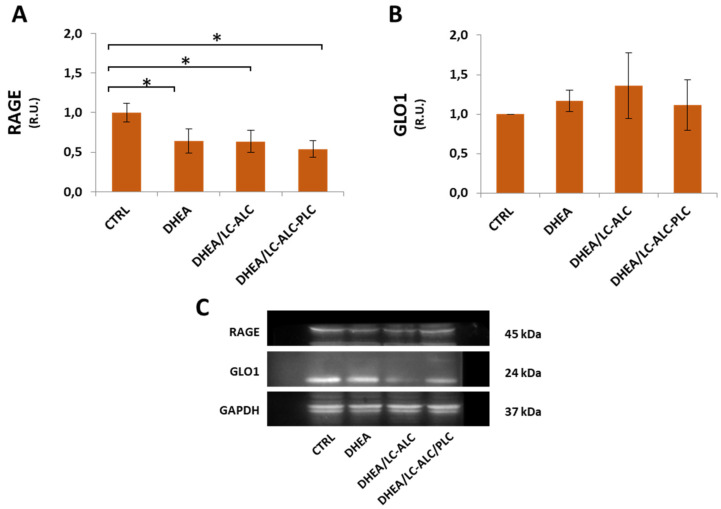
Western blot analysis of RAGE (**A**) and GLO1 (**B**) and representative images (**C**). Three mice per experimental group were employed. Experiments were done in triplicate. Data obtained from n = 9 observations are presented as means ± SEM of densitometric analysis of immunoreactive bands normalized to internal reference protein (glyceraldehyde-3-phosphate dehydrogenase, GAPDH). * *p* < 0.01 vs. CTRL, *t*-test.

**Table 1 biomedicines-11-00374-t001:** Morphometric measurements on the thickness of the endometrium, epithelium and stroma of mouse uteri. Values are expressed as mean ± SD, Differences evaluated by ANOVA followed by Tukey’s test. Superscripts indicate a significant difference (*p* < 0.05).

	Control (µm)	DHEA (µm)	DHEA/LC-ALC (µm)	DHEA/LC-ALC-PLC (µm)
Endometrium	141.615 ± 13.266 ^a^	345.584 ± 17.199 ^b^	237.225 ± 40.884 ^c^	171.683 ± 8.181 ^a,c^
Epithelium	12.464 ± 1.685 ^a^	25.119 ± 2.370 ^b^	19.435 ± 2.582 ^a,b^	13.747 ± 1.494 ^a^
Stroma	129.864 ± 13.419 ^a^	321.645 ± 14.989 ^b^	209.886 ± 38.386 ^a^	159.004 ± 7.373 ^a^

## Data Availability

Data supporting the findings of this study are available from the corresponding author upon reasonable request.

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
