# Peer review of "Modulating Morphological and Redox/Glycative Alterations in the PCOS Uterus: Effects of Carnitines in PCOS Mice"

_biomedicines, 2023, doi:10.3390/biomedicines11020374_

Round 1

Reviewer 1 Report

Cellular senesence is an age-decline of reproductive function including ovarian and uterine dysfunction. Therefore, paper by Secomandi et al.,Human Reproduction Update, Vol.28, No.2, pp. 172–189, 2022 should be cited and discussed.

Author Response

Comment- Cellular senesence is an age-decline of reproductive function including ovarian and uterine dysfunction. Therefore, paper by Secomandi et al.,Human Reproduction Update, Vol.28, No.2, pp. 172–189, 2022 should be cited and discussed.

Response-We thank you for having appreciated our research work and for your suggestions. As you proposed we cited the manuscript below on page 17 lines 489,493  with reference to the following comments:

Inflammation may impair physiological clearance of senescent decidual cells and turn into a chronic process resulting in defective functionality.

Overall based on this observations PCOS uterus phenotype may resembles aging uterine phenotype as described in preclinical models .

We hope that these answers could properly address your observations.

Reviewer 2 Report

The authors present the results of animal studies focused on the effects of different carnitine formulations on the morphological and molecular characteristics of the PCOS uterus. The topic is of interest and the results look promising as a background for the potential practical use of carnitine in patients with PCOS in the future.

The general estimation of the manuscript is positive, nevertheless, there are some issues to clarify.

1. The authors used a DHEA-induced mouse PCOS model, but do not provide the data or references regarding validation of this model, as well as the reasons to use this exactly animal PCOS model.

2. The Methods section. The description of statistical analysis for the sub-section 2.7. looks incomplete. 

3. Concerning the images (Fig 2,4). A scale indicating the magnification should be presented on each image.  The scales in the lower right corners of images are unreadable.  

4. Fig. 5,8,10:  t-test is appropriate if the distribution is normal, but that  looks doubtful because of few observations (data from 3 mice, in triplicates).

Author Response

Comment 1. The authors used a DHEA-induced mouse PCOS model, but do not provide the data or references regarding validation of this model, as well as the reasons to use this exactly animal PCOS model.

Response: We answered this comment on page 2 lines 91-92.

Comment 2. The Methods section. The description of statistical analysis for the sub-section 2.7. looks incomplete. 

Response: To address this issue we added the statistical analysis section (2.8) on page 5 lines 215-221.

Comment 3. Concerning the images (Fig 2,4). A scale indicating the magnification should be presented on each image.  The scales in the lower right corners of images are unreadable.  

Response: The scales were corrected in Figure 2,4.

Comment 4. Fig. 5,8,10:  t-test is appropriate if the distribution is normal, but that  looks doubtful because of few observations (data from 3 mice, in triplicates).

Response: This issue was addressed by the information in the statistical section and in the legend of figures 5,8,10.

We thank you for having appreciated our research work and for your suggestions and hope that these answers could properly address your observations.